# Application Prospect of Induced Pluripotent Stem Cells in Organoids and Cell Therapy

**DOI:** 10.3390/ijms25052680

**Published:** 2024-02-26

**Authors:** Teng Zhang, Cheng Qian, Mengyao Song, Yu Tang, Yueke Zhou, Guanglu Dong, Qiuhong Shen, Wenxing Chen, Aiyun Wang, Sanbing Shen, Yang Zhao, Yin Lu

**Affiliations:** 1Jiangsu Joint International Research Laboratory of Chinese Medicine and Regenerative Medicine, Nanjing University of Chinese Medicine, Nanjing 210023, China; 20210757@njucm.edu.cn (T.Z.); qiancheng@njucm.edu.cn (C.Q.); 20200662@njucm.edu.cn (M.S.); 20200695@njucm.edu.cn (Y.T.); zyk9952445@163.com (Y.Z.); chenwx@njucm.edu.cn (W.C.); wangaiyun@njucm.edu.cn (A.W.); 2Jiangsu Key Laboratory for Pharmacology and Safety Evaluation of Chinese Materia Medica, Nanjing University of Chinese Medicine, Nanjing 210023, China; dongguanglu0630@126.com (G.D.); sqh@njucm.edu.cn (Q.S.); 3Department of Biochemistry and Molecular Biology, School of Medicine & Holistic Integrative Medicine, Nanjing University of Chinese Medicine, Nanjing 210023, China; 4Regenerative Medicine Institute, School of Medicine, University of Galway, H91 W2TY Galway, Ireland; sanbing.shen@universityofgalway.ie

**Keywords:** induced pluripotent stem cells, reprogramming, disease modeling, cell therapy, organoids, regenerative medicine

## Abstract

Since its inception, induced pluripotent stem cell (iPSC) technology has been hailed as a powerful tool for comprehending disease etiology and advancing drug screening across various domains. While earlier iPSC-based disease modeling and drug assessment primarily operated at the cellular level, recent years have witnessed a significant shift towards organoid-based investigations. Organoids derived from iPSCs offer distinct advantages, particularly in enabling the observation of disease progression and drug metabolism in an in vivo-like environment, surpassing the capabilities of iPSC-derived cells. Furthermore, iPSC-based cell therapy has emerged as a focal point of clinical interest. In this review, we provide an extensive overview of non-integrative reprogramming methods that have evolved since the inception of iPSC technology. We also deliver a comprehensive examination of iPSC-derived organoids, spanning the realms of the nervous system, cardiovascular system, and oncology, as well as systematically elucidate recent advancements in iPSC-related cell therapies.

## 1. Background

The remarkable ability of pluripotent stem cells (PSCs) to self-renew and develop into any type of somatic cell is what sets them apart. These cells can originate from embryonic cells or adult somatic cells. Ethical constraints have limited the clinical application of embryonic stem cells (ESCs), which are derived from embryos [1]. This limitation has led to the emergence of iPSCs, which share similarities with ESCs but bypass ethical concerns by utilizing patient-specific somatic cells. The inception of human iPSC technology in 2007 marked a pivotal moment [2]. While initial reprogramming methods suffered from low efficiency, subsequent refinements have enabled robust and efficient iPSC generation without the genomic integration of transgenes. Conventional iPSC reprogramming often relies on integrating viral vectors, carrying the risk of insertional mutagenesis. To mitigate this risk, non-integrative reprogramming methods, such as adenovirus, endai virus (SeV), and protein-based approaches, have gained favor among researchers.

Organoids, self-organizing cell aggregates derived from human stem cells, closely mimic the cell types and organizational properties of native organs [3]. iPSCs play a pivotal role in organoid generation, yielding diverse and physiologically relevant models adaptable to normal and disease conditions. The inability of traditional two-dimensional (2D) cell cultures to replicate the cellular processes found in tissues is due to their lack of cellular variety, hierarchical structure, and cell-matrix or cell-cell interactions [4]. iPSC-derived three-dimensional (3D) organoids, with their multicellular and multifunctional nature, replicate organ-like properties. Moreover, researchers can manipulate the 3D culture system by adjusting growth factors and differentiation protocols to create a spectrum of organoids mirroring normal or disease states [4]. Thus, iPSCs offer a versatile platform for organoid development.

In this paper, we explore iPSC reprogramming techniques, emphasizing iPSC-derived organoids and their potential applications in disease therapy. These advancements hold promise in advancing our understanding of disease mechanisms and facilitating drug screening and therapy development.

## 2. Nonintegrating Way of iPSC Reprogramming

Since the successful isolation of human ESCs in 1998, significant efforts have been directed towards differentiating ESCs into mature tissues, particularly for disease modeling and the long-term goal of allogeneic cell therapy [5]. However, prior to the development of gene targeting techniques, genome editing in human ESCs was plagued by remarkably low efficiency, and establishing clear genotype-phenotype correlations within ESC-based models was challenging [6]. Consequently, obtaining specific mutations in ESCs proved to be a formidable task. More importantly, the advent of reprogramming technology marked a pivotal shift, facilitating the utilization of disease-specific iPSCs within established directed differentiation protocols. This breakthrough overcame the limitations associated with disease modeling in ESCs [5].

While iPSCs were officially reported in 2006, the foundation for reprogramming had been laid long before. iPSCs offer diverse sources compared to other PSCs with various reprogramming methods at our disposal [7]. For example, PSCs have been generated from human and mouse somatic cells by expressing transcription factors such as Oct4, Sox2, and Klf4 by viruses. Nevertheless, a significant disadvantage of this strategy is the employment of viruses that integrate into the genome, which increases the likelihood of tumor formation since viral transgenes may reactivate [8]. Lentiviral vectors, another gene transfer method, integrate the transgene into the host chromosome, theoretically ensuring long-term transgene expression. Nevertheless, this approach may yield unstable transgene expression or unexpected side effects, such as overexpression or inactivation of unrelated genes [9]. To circumvent potential insertional mutagenesis, transient expression via non-integrating vectors has emerged as a promising alternative [10]. The subsequent sections briefly introduce several non-integrating reprogramming methods.

### 2.1. Adenovirus

Adenoviral vectors are widely employed in both experimental research and clinical trials due to their ability to efficiently transfer genes into various somatic, stem, and cancer cell types. iPSCs can be generated from both mouse and human somatic cells using adenoviral vectors, enabling transient and robust expression of exogenous genes without genomic integration [8,11].

### 2.2. SeV

Another widely employed reprogramming method utilizes SeV-derived particles [12]. SeV, an RNA virus primarily affecting respiratory tissues in rodents, readily infects various murine and human cells by binding to sialic acid residues expressed on target cells [12]. Since SeV does not replicate during the nuclear phase, there’s no chance of host genome modification or gene silencing [13]. Consequently, SeV is extensively used as a vector in stem cell research, enabling efficient reprogramming of cells. Compared to other integrative strategies, SeV has demonstrated high reliability, superior efficiency in generating iPSCs free from viral contamination, and reduced workload [13]. For instance, SeVdp-302L, delivering a single-stranded RNA genome with multiple transgenes into somatic cells for iPSC generation, ensures stable transgene expression and includes an auto-erasable feature responsive to stem cell-specific microRNA-302 [14].

### 2.3. Protein

The notion of using proteins as a means to induce reprogramming for generating footprint-free iPSCs was proposed over a decade ago [15]. Cho’s pioneering study showed that pluripotent state reprogramming may be achieved without forcing the expression of ectopic transgenes with a single transfer of proteins obtained from embryonic stem cells into predominantly cultivated adult mouse fibroblasts. This discovery unveiled the remarkable potential of protein-iPSCs, showcasing their ability to undergo in vivo differentiation, as evidenced by well-differentiated teratoma formation. Furthermore, these protein-iPSCs displayed developmental potential by enabling the generation of chimeric mice and participating in tetraploid blastocyst complementation experiments [16]. Despite its advantages, it’s worth noting that synthesizing large quantities of bioactive proteins capable of efficiently crossing the plasma membrane has presented technical challenges.

### 2.4. mRNA/miRNA

mRNA reprogramming is now widely adopted due to its transient vector characteristics, which enable rapid and efficient iPSC generation. It offers precise control over reprogramming factors, including dosing, stoichiometry, and timing [17]. In late 2010, Warren published a seminal study demonstrating iPSC induction via mRNA transfection is feasible [17,18]. Compared to traditional mRNA reprogramming, modified mRNA (modRNA) addresses key limitations, such as the short persistence of protein expression after mRNA transfection and rapid cytoplasmic degradation. ModRNA exhibits improved stability and reduced immunogenicity [19]. One notable challenge in realizing mRNA reprogramming is the potent activation of broadly expressed antiviral defense pathways in mammalian cells upon synthetic mRNA delivery. This activation suppresses protein translation from exogenous transcripts and triggers cytotoxic and cytostatic responses detrimental to reprogramming [17]. Consequently, extensive research is required to overcome these hurdles.

### 2.5. PiggyBac

The PiggyBac (PB) transposon system is recognized for its demonstrated effectiveness in the genomic engineering of mammalian cells, playing a pivotal role in various preclinical applications. This versatile tool has facilitated gene discovery, multiplexed genome modification, animal transgenesis, and in vivo gene transfer, ensuring sustained gene expression in animals. Additionally, it has been instrumental in genetically modifying clinically relevant cell types, including iPSCs and human T lymphocytes [20]. One remarkable advantage of this non-viral approach is its ability to simplify and reduce the cost of producing the four-factor vector [20]. Notably, PB also allows for precise excision of reprogramming factors from iPSCs by re-expressing the PB transposase when introduced on a transposon [20]. Woltjen uncovers the successful and effective reprogramming of human and mouse embryonic fibroblasts with PB transposition-delivered doxycycline-inducible transcription factors [21]. More recently, numerous researchers have leveraged CRISPR/Cas9 technology to edit the PB transposon, enabling iPSC reprogramming [22,23,24].

### 2.6. Episomal Plasmids

A decade ago, lasmid vectors encoding Oct3/4, Sox2, Klf4, L-MYC, and Lin28 have been successfully used to induce human iPSCs from peripheral blood mononuclear cells (PBMCs) [25]. Remarkably, they could establish hundreds of iPSC lines from as few as 1 × 10^6^ PBMCs isolated from donors ranging in age from their 20s to their 60s [25]. Additionally, combing episomal plasmids with the small chemical sodium butyrate also enhances the effectiveness of reprogramming when generating integration-free human iPSCs from differentiated adult fibroblasts [26]. In conclusion, this attractive footprint-free method is simple to produce iPSCs from a single transfection in a variety of cell types. It shows promise for upcoming therapeutic applications and is well-suited for the production of iPSCs tailored to individual patients. Plasmids reprogramming has proven effective in generating footprint-free iPSCs, with the main limitation being the need for modified cell culture methods to achieve acceptable reprogramming efficiency in fibroblasts. The efficiency, difficulty, and carrier removal of the above reprogramming methods are summarized, as shown in Table 1.

## 3. Construction of Organoids Based on iPSC Technology

Human organoids, 3D cellular structures cultured from adult or PSCs, faithfully recapitulate key morphological, functional, and transcriptomic aspects of human organs [32,33,34,35]. These organoids are essential tools for understanding the mechanisms behind a variety of diseases, regardless of whether they were created to carry mutations that cause the disease or were taken directly from patient cells [32,36,37]. They offer a more precise reflection of the intricate cellular ecosystems found in complex tissues, making them ideal for disease modeling and drug screening.

Traditionally, translating drug trial results from experimental animal models to humans has been fraught with challenges due to species-specific variations in biological responses. A small number of new medications that the US Food and Drug Administration approves for clinical use each year is proof that this mismatch frequently leads to significant failure rates [38]. However, recent advancements in scientific methods, particularly the development of 3D organoid cultures and 3D-printed scaffolds, are revolutionizing this landscape [39]. This technological breakthrough in 3D organoid models holds great promise, enabling a comprehensive exploration of human disease complexities and facilitating in-depth investigations into disease pathogenesis [39]. However, with the expanding applications of iPSC technology, iPSC-derived organoid cultures are emerging as a dominant trend. These organoids are invaluable for mimicking a wide range of human maladies, including infectious diseases, genetic abnormalities, and cancer, since they realistically replicate the complex cellular ecosystems of complex tissues, in conjunction with human-rodent chimeras [39]. Particularly, patient-derived iPSC-derived organoids that can truly reflect the pathological characteristics of each individual, avoid possible adverse reactions, and make personalized treatment more accurate.

### 3.1. Derivation of Neural Organoids

Considering the human brain is complex, it’s difficult to recreate in mice, which makes an in vitro human brain model necessary to investigate many brain illnesses in model organisms. Consequently, a significant body of research has been dedicated to developing a 3D organoid culture system derived from human iPSCs.

The innate immune cells of the central nervous system, known as microglia, are essential for tissue homeostasis and neurodevelopment [40,41,42,43,44,45,46,47,48]. They are responsible for tasks such as phagocytizing apoptotic cells, pruning redundant synapses, and regulating neurogenesis and axonal growth [49]. Chronic inflammation is a key characteristic of neurological disorders, and modeling neuroinflammation-related diseases cannot be done without microglia. A growing amount of research has also demonstrated the connection between microglial dysfunction and a variety of neurological conditions, such as traumatic brain injury, Alzheimer’s disease (AD), and schizophrenia [49,50]. The lack of microglia, which presents the capability to reorganize brain networks and engulf dead cells and detritus, is a significant shortcoming for the neural organoid systems that are already in use. There are two ways to obtain microglia-containing organoids, one is to incorporate homologous microglia into the completed organoids, and the other is to generate microglia-containing brain organoids directly based on iPSCs. The former physically attaches microglia to the surface of organoids by means of co-culture. For instance, in the process of midbrain organoid differentiation, a coculture medium containing 186,000 freshly harvested macrophage precursor cells per organoid was substituted for the culture medium of midbrain organoids and assembloids starts on day 15 of dopaminergic (DA) differentiation [51]. By contrast, the latter obtained organoid models containing microglia by co-culturing iPSC-derived progenitors and progenitors of macrophages. According to recent studies, iPSCs taken from healthy individuals were used to generate primitive neural and macrophage progenitor cells. These progenitors were then co-cultured with 7000 neural and 3000 macrophage progenitors to create brain organoids that contained microglia [49]. Although the above studies actually obtained microglia-containing organoids, presently, iPSC-derived neural organoids containing microglia are relatively few, and there’s still a certain gap between these organoids and the real brain in terms of the distribution and quantity of microglia. Microglia only physically cover the surface of the organoids and do not really recapitulate the physiological processes in the body, which needs to be further addressed in the future.

Additionally, amyotrophic lateral sclerosis (ALS), Parkinson’s disease (PD), and AD can all be effectively modeled by iPSC-derived organoids. While challenges related to differentiation efficiency, maturation periods, and recapitulating sporadic or late-onset disease phenotypes persist, recent research suggests that iPSC-based organoids hold significant promise for advancing research and discovery in the field of neurodegenerative diseases [52,53,54,55,56,57].

### 3.2. Derivation of Liver Organoids

The evolving comprehension of liver organogenesis mechanisms has paved the way for innovative strategies in personalized liver disease modeling and treatment through human iPSC-derived hepatic sources and stromal cellular compositions [58]. The common method of differentiation of iPSCs into hepatocytes and organoids is shown in Figure 1. A multitude of researchers are currently focused on developing various liver organoids tailored to address numerous liver conditions [59,60,61,62], including fatty liver [63], hepatic steatosis [64], hepatitis, liver fibrosis [65], and irreversibly damaged liver.

While it’s interesting to obtain liver organoids from iPSCs, it’s also important to improve existing methods. A strategy of direct dialysis-based media conditioning was employed to induce the differentiation of human liver organoids [71]. By optimizing the accumulation of growth factors, they significantly enhanced hepatic differentiation of human iPSCs at high cell densities. The resultant human liver organoids showed hepatobiliary physiology and hepatic indicators that were on par with or superior to those that had been differentiated at lower cell densities in suspension cultures with regular daily media replacement [71].

Indeed, up to a third of adult liver transplants and 70% of pediatric liver transplants are caused by problems with the biliary system, which moves bile from the liver to the duodenum [72]. Therefore, it’s urgent to obtain organoids with hepatobiliary characteristics. The process of causing human iPSCs to develop into 3D hepatobiliary organoids has advanced. Researchers achieved this by supplementing hepatic differentiation medium with 25% mTeSR^TM^ (STEMCELL Technologies, Cat:85850, Vancouver, Canada) culture medium and 10% cholesterol+ MIX to induce endodermal and mesodermal commitment, leading to the formation of mature hepatobiliary organoids [73]. These organoids demonstrated appropriate secretion abilities (albumin and urea) and drug metabolic capacity (CYP3A4 activity and inducibility). Furthermore, they exhibited remarkable survivability in immune-deficient mice for over 8 weeks and proved valuable for in vitro studies investigating the molecular mechanisms of liver development, holding substantial potential for liver disease therapy [73].

Additionally, maturity has always been one of the major factors that hinder the application of organoids. Thus, to obtain mature liver organoids, the researchers have built liver organoids with a rich vascular structure. Takebe demonstrated a groundbreaking achievement in generating vascularized and functional human liver tissue from human iPSCs through the transplantation of in vitro-created liver buds. Certain hepatocytes were autonomously arranged into 3D iPSC liver buds by mimicking the organogenetic interactions between endothelial cells and mesenchymal cells [74]. By adhering to host capillaries, these transplanted human iPSC liver buds successfully established functional vasculatures. Without the need for recipient liver replacement, the highly metabolic iPSC-derived tissue carried out liver-specific tasks such as protein synthesis and human-specific drug metabolism [74]. Additionally, in multi-organ systems generated from iPSCs, it has been demonstrated that intertissue communication promotes organ maturation [75]. For example, in co-cultivation, liver and islet organoids generated from iPSCs consistently exhibit robust growth and tissue-specific function [76]. In addition, it has recently been demonstrated that human liver-pancreatic islet axis recapitulation in both healthy and diseased conditions is possible in terms of using microfluidic multi-organ systems [77]. The technology allows for the 3D co-culture of human islet and liver organoids produced from iPSCs for up to 30 days under circulatory perfusion conditions. It’s made up of two divided zones connected by a network of microchannels [77]. By offering precise biomechanical cues, biochemical signals, and organ-organ interactions, organoids-on-a-chip devices provide a regulated tissue microenvironment [78,79]. Consequently, it’s possible to produce a large number of organoids with highly similar composition structures, indicating that the differences between the organoids obtained from the same batch production are smaller, by utilizing the controlled culture conditions of chip technology, the features of diversified stimulation, and continuous parameter readout to enhance the mutual communication between iPSC-derived organoids. However, the differentiation parameters for providing individual organoids in multi-organ chips have not been reported.

The field will still confront a number of obstacles in the future, despite these impressive advances. These include the following: (1) creating strategies to modify the intrinsic cellular composition, distribution, and proportion within liver organoids; (2) identifying their unique functions in the development and pathophysiology of the liver; (3) clarifying the biochemical characteristics of extracellular matrices (ECM) specific to the liver; and (4) incorporating ECM hydrogels or microparticles that are clinically compatible. It will be essential to address these issues if liver organoids’ physiological properties are to be preserved throughout modeling [58].

### 3.3. Derivation of Cardiac Organoids

Heart disease, a leading global cause of death, imposes a substantial burden on healthcare systems. Given its diverse manifestations and high mortality rates, there’s an urgent need for a deeper understanding of its various mechanisms.

The development of human heart organoid models for cardiovascular disease research has been slow and trails much behind that of other organs, despite the significance of comprehending human cardiovascular illnesses for treatment and prevention. There’s a great need to close this knowledge and technical gap because creating more accurate in vitro human heart models would help scientists and clinicians better understand both healthy and sick states for research and translational applications (Figure 2).

Hence, many methods and techniques for generating cardiac organoids have emerged. For instance, human cardiac fibroblasts (HCFs), human cardiac microvascular endothelial cells (HCMECs), and human iPSC-derived cardiomyocytes (iPSC-CMs) are mixed in a 3:5:2 cell ratio. Then, in a 60-well plate, a 20 µL cell suspension containing about 100,000 cells was pipetted in each well and organoids are harvested three days later [80]. Furthermore, the potential of a microgel-based biphasic (MB) bioink is disclosed, which exhibits shear-thinning and self-healing behavior, both as an excellent bioink and a suspension medium for embedded 3D printing. When human iPSCs were encapsulated in the MB bioink, substantial stem cell proliferation and cardiac differentiation were able to generate cardiac tissues and organoids [81].

However, methods for generating cardiac organoids with native heart morphology based on iPSC technology are still lacking. The WNT signaling pathway, an important signaling pathway that regulates cardiomyocyte differentiation and cardiac development [82,83,84,85], can alleviate the above problems. Stated differently, during the routine manufacturing and differentiation of cardiac organoids, WNT inhibitors and activators are added at specific times, resulting in the generation of significant heart-like structures with respect to structure, organization, function, the complexity of cardiac cell types, ECM composition, and vascularization. For instance, on the 0 day of organoid differentiation, CHIR99021 (activator of WNT signaling pathway) was added for 24 h, and then the old medium in the pore plate was removed and the new medium was added. On the second day of organoid differentiation, WNT-C59 (nhibitors of the WNT signaling pathway) was added for 48 h, and then the organoids with heart-like structures could be obtained by continuing culture with normal medium [86]. As an alternative, mesodermal WNT-BMP signal axis targeting can potentially impart a hollow structure to heart organoids. Furthermore, studies reveal that the high level of WNT signaling during mesoderm induction promotes cavity growth in the later cardiac mesoderm stage [87], yet iPSC-derived cardiac organoids have not yet used this method. Organs produced from iPSCs therefore have greater development potential.

Organoids can mimic many physiological aspects of tissue development in vitro, but they usually lack the diverse multilineage progenitors necessary for the development of complex organs like the heart, as well as the cooperative paracrine interactions between adjacent tissues that characterize embryogenesis. It was thus shown that paracrine communication between tissues in close proximity helps to the determination of their respective cell fates and maturation into functioning organs. Researchers also disclosed a human multilineage iPSC-derived organoid. To establish concentric cardio-pulmonary micro-Tissues, for instance, a novel stepwise strategy is used to direct the simultaneous induction of both mesoderm-derived cardiac and endoderm-derived lung epithelial lineages within a single differentiation of human iPSCs via temporal specific tuning of WNT and nodal signaling in the absence of exogenous growth factors. This organoid quickly matured alveoli in the presence of cardiac accompaniment [88]. There has also been the production of another multi-lineage iPSC-derived organoid that, over extended periods of time, recapitulates the co-development, differentiation, and maturation of two distinct tissues (the gut and the heart) and shows enhanced physiological maturation of cardiac tissue, notably of atrial/nodal cardiomyocytes [89]. Multilineage organoids are a natural next step towards more physiological in vitro models of human development, but more work needs to be done to expand the opportunities for simulating complicated multi-organ disorders or disorders of tissue morphogenesis or maturation ex vivo.

### 3.4. Derivation of Cancer Organoids

Model systems are used in both translational and basic cancer research to replicate the malignant state at the molecular, cellular, tissue, organ, and organism levels. The scientific community’s interest in creating patient-derived cancer models has been reignited due to growing worries about the low rate at which fundamental research findings are applied and the fact that cancer is a far more complex disease than previously thought. In addition to current developments in genome editing, xenotransplantation, bioengineering, and the processing and culture of human tissue, iPSCs provide a number of novel possibilities for the research of human cancer. iPSC-derived cancer cells may offer a fresh approach to the field of cancer research. During the process of reprogramming, these cancer cells may take on characteristics of cancer stem cells. Alternatively, the differentiation of cancer-iPSCs by teratoma development or organoid culturing may imitate the tumorigenic process. The use of the cancer-iPSC model could shed light on a few molecular processes linked to the advancement of cancer [90].

With the change of modern lifestyle, the incidence of colitis has gradually increased, and the prone population is very wide, difficult to cure, and the cancer rate is high, so it’s listed as one of the modern difficult diseases by the World Health Organization [91]. Long-term colitis and adenomatous polyposis can lead to contractions, perforations, and even bowel cancer [92,93,94,95,96,97,98,99,100]. Colon cancer organoids can be derived based on iPSC technology [101]. Crespo tried to create a successful method for obtaining differentiated human iPSCs that would allow for the production of colonic organoids (Cos), which would be used to simulate diseases of the large intestine in humans [102]. In this study, extensive gene and immunohistochemical profiling confirmed that the derived Cos represent the colon rather than the small intestine, the author applied this strategy to iPSCs derived from patients with familial adenomatous polyposis harboring germline mutations in the WNT-signaling-pathway-regulator gene encoding adenomatous polyposis coli (APC), and Cos exhibits enhanced WNT activity and increased epithelial cell proliferation, which was used as a platform for colorectal cancer model and drug testing [102]. The iPSC-derived colon cancer model is highly similar to the traditional colon cancer model. iPSC-derived organoids, on the other hand, differ from patient-derived organoids in the case of gastrointestinal cancer. There are a few obvious distinctions between the two: The first is that iPSC-derived organoids require a lot longer to mature than organoids obtained from patients; the second is that iPSC-derived organoids have a far more complicated maturation process with numerous layers and stroma [103]. However, the stromal compartments seen in iPSC-derived organoids may be very helpful for researching cancer, as it has been observed that pancreatic cancer cells’ stromal compartments do not function normally. Because stromal compartments are present in iPSC-derived organoids, it’s feasible to monitor and control intracellular communication to influence probable illness causes or more effectively administer chemotherapy and other cancer-targeting drugs [103]. Furthermore, genetically modified iPSC-derived organoids are helpful for studying how the microenvironment of cancer stem cells is regulated, which is important for determining how cancer tissues, normal cell organoids, and cancer organoids differ from cancerous regions [104]. iPSC-derived organoids are also different from organoids produced by tissue stem cells. iPSC-derived organoids are produced by sequential exposure to a series of signaling cues, which can provide possible mechanisms for the development and trajectory of disease. It has been accepted that iPSC-derived organoids are diverse. Tissue stem cells are derived from both normal and diseased tissues, providing the possibility to study specific tissues [105]. For instance, it’s interesting to use CRISPR-Cas9 technology to induce genetic mutations in colon organoids of normal tissue origin to study the genetic profile of colorectal tumors [106]. Moreover, organoids derived from tissue stem cells also have the potential for long-term expansion in vitro [107].

Retinoblastoma is a childhood cancer of the developing retina that starts with biallelic inactivation of the retinoblastoma gene 1 (RB1) gene [108,109]. Children with germline mutations in RB1 have an increased risk of developing retinoblastoma and other cancers later in life. Early on, several researchers tried to use R1 gene mutations to imitate retinoblastoma in mice, but they were unable to cause retinal tumors in Rb^+/−^ mice. The subsequent conditional inactivation of both copies of R1 in the embryonic murine retina similarly failed to create retinoblastoma, since species-specific intrinsic genetic redundancy and compensation among Rb family members prevent retinoblastoma in mice [108]. Consequently, a few researchers created iPSCs from fifteen individuals who had germline RB1 mutations or deletions. Multiple clones from each iPSC line were then confirmed to retain the germline mutation and exposed to molecular profiling [110]. Subsequently, representative clones of every participant were created, developed into retinal organoids, separated, and injected intraocularly into the eyes of immunocompromised mice to track the development of tumors. This approach offers a fresh model for researching malignancies. Retinoblastoma, on the other hand, can further damage the eye structure, resulting in conjunctival congestion and corneal damage. iPSCs can also self-organize into autonomous multizone ectoderm of ocular cells, and then form functional corneal epithelial cell sheets [111]. Corneal epithelial cells obtained in this way successfully ameliorated corneal barrier dysfunction and can also ameliorate corneal damage caused by retinoblastoma.

One common reason of epithelial malignancies is the mutant Kirsten rats sarcoma viral oncogene homolog gene (KRAS). However, little is known on the molecular alterations that take place in epithelial cells immediately following the activation of oncogenic KRAS. In addition to patient samples and the genetically engineered mouse model, the development of organoid systems from primary mouse and human iPSC-derived lung epithelial cells to simulate lung adenocarcinoma has become a reality and demonstrated the utility of in vitro organoid approaches for uncovering the early consequences of oncogenic KRAS expression [112].

A bottleneck in the process of generating organoids from iPSCs is that traditional differentiation methods can only differentiate iPSCs into a specific type of cell, rather than the multiple cell types that make up tissues and organs. In order to achieve organ-level cellular variety, current techniques for creating cerebral, renal, retinal, and other organoids frequently call for extended culture durations, ranging from a few weeks to several months [113]. By simultaneously co-differentiating human iPSCs into distinct cell types via the forced overexpression of transcription factors, independent of culture-media composition, it’s possible to generate patterned organoids and bio-printed tissues with controlled composition and organization, but it will still require a significant financial and time investment [113].

## 4. Screening Drugs and Toxicity Detection of iPSC-Derived Organoids

Traditional iPSC-derived 2D cell culture systems have been shown to be an incredibly useful resource, yielding vital insights and lowering the cost of disease modeling techniques [114]. However, the primary drawback of iPSC-derived 2D cell lines is that they are unable to replicate the cellular functions found in tissues due to their lack of cellular variety, dimensionality, hierarchical structure, and cell–cell or cell–matrix interactions [115]. Scientists contend that the complexity of human organs is not captured by 2D models, necessitating the development of more medically accurate models [115]. Over the past decade, iPSC technology has advanced significantly. When coupled with recent breakthroughs in gene editing and 3D organoid development, iPSC-based platforms have gained enhanced capabilities for drug screening. As shown in Figure 3, this approach is gaining widespread popularity, driven by the growing interest in phenotypic screening and the distinct advantages offered by human iPSCs compared to traditional cellular screening methods.

### 4.1. Neural Organoids

Owing to the fact that iPSCs are able to differentiate into 3D organoids, a model that authentically simulates tissue and organ-level disease pathophysiology for drug screening can be created that nearly mimics the complexity of the brain [116,117,118,119,120,121]. Compared to conventional 2D cultures, organoids display a series of advantages. Their cellular makeup is almost physiologic, and they can grow large in culture without losing genetic integrity, which can enable them to be useful for high-throughput testing. Given that 3D organoids have a morphology and structure more similar to genuine tissues than iPSC-derived cells or 2D cultures, they can more accurately represent disease and physiological states at the tissue and organ level. As such, for the purposes of drug screening and disease modeling, iPSC-derived 3D organoids perform better than iPSC-derived 2D cells.

Therefore, a variety of iPSC-derived neural organoids are used as platforms to screen for effective drugs. To speed up the search for therapeutics for diseases including schizophrenia, autism, and epilepsy, researchers tend to implement robotics and artificial intelligence-based phenotypic screening on iPSC-derived neural organoids. These researchers created motor neurons from iPSCs derived from familial ALS patients, which displayed cytosolic aggregates resembling those observed in ALS patients’ postmortem tissue [122]. Another example is that the researchers evaluated the therapeutic effects of four chemical compounds on ALS and identified anacardic acid, a histone acetyltransferase inhibitor, as a potential rescue agent for the abnormal ALS motor neuron phenotype [123]. Recently, ropinirole, a novel medication, has just started a phase 1/2a clinical trial. This medication prolongs ALS patients’ disease-progression-free survival by successfully reducing the expression of neurofilament light chain protein in the cerebrospinal fluid of ALS patients. An in vitro model of cytoplasmic aggregates produced by neurons originating from iPSCs was used to choose this medication [124]. This strongly implies that medication candidates can be successfully screened using ALS disease models produced from iPSCs.

However, despite the versatility of iPSC-derived neural organoids, their effective application in large-scale drug screening still faces challenges of heterogeneity, scalability, repeatability, and maturity [122]. Effectively solving these problems is the key to the current breakthrough. For instance, a differentiation technique from iPSC-derived expandable primitive neural stem cells is established, enabling the rapid generation of simplified brain organoids within two weeks, consisting of mature neurons and astrocytes [125]. This addresses the problem of organoid scalability. Immunochemistry and single-cell studies confirmed that the cytoarchitecture and transcriptional characteristics of these 3D cultures were uniform [125]. Taken together, more efforts are needed to make breakthroughs in organoid maturation and heterogeneity.

Additionally, recent advances in tissue stem cell research have also been significant. Hendriks D and co-workers successfully used fetal brain tissue to self-organize and form complex brain organoids in vitro [107]. This organoid contained a variety of cell types in brain tissue, including neurons, neural stem cells, and various progenitor cells, and was highly heterogeneous [107]. Because of this, it had the characteristics of long-term expansion and better organizational similarity. Compared with this kind of organoid, iPSC-derived organoids exhibited worse heterogeneity. If researchers want to obtain organoids with good heterogeneity from iPSCs, more complex signaling interventions are required in the process. Nevertheless, the sources of iPSCs are extensive and the ways to obtain them are varied. For the above brain organoids, the source may only be fetal brain tissue, which is very limited. However, it’s still a great innovation and a major technological breakthrough.

### 4.2. Liver Organoids

Developing more effective and repeatable procedures to generate liver organoids that can be utilized for drug screening has attracted increasing attention ever since the first reports of creating hepatocyte-like cells from iPSCs [126,127,128,129].

Currently used to prepare liver organoids, the classic Matrigel dome approach lacks precise microenvironment control and produces liver organoids with significant variances [130,131]. This is not conducive to the screening of drugs that have a therapeutic effect on liver disease, but research has been done on this problem. For instance, a culture technique that enables fine control over the size, density, and mass of individual liver organoids has been reported recently, along with a method for manufacturing high throughput liver organoids [130]. Without the use of Matrigel, human iPSCs are first developed into human foregut stem cells (hFSCs), which are subsequently further differentiated into liver organoids inside agarose-based microfabricated hexagonal arrays [130]. This liver organoid serves as a preclinical model to evaluate the liver’s response to acute hepatotoxicity caused by acetaminophen (APAP) exposure [130]. It’s possible to create a novel liver organoid that closely mimics important aspects of human liver development during the fetal stage by using human iPSC technology and micropatterning techniques. By virtue of this technique, bioengineered fetal liver organoids with deterministic size and position in a multi-well plate and consistent morphology may be created in a repeatable, high-throughput manner [132]. This organoid effectively demonstrates the hepatotoxicity of APAP and its metabolism in the liver. In addition, Shinozawa developed a reproducible method to generate human liver organoids from storable foregut progenitors derived from PSCs lines. These organoids were utilized to examine the toxicity of drugs since they had structures resembling functional bile canaliculi. A novel organoid-based assay was developed by the researchers, which demonstrated strong predictive values for 238 commercially available medications based on multiplexed readouts evaluating survivability, cholestatic, and/or mitochondrial toxicity [133].

Nevertheless, low cytochrome P450 (CYP450) enzyme expression and activity is responsible for the diminished drug metabolic function of the current in vitro liver models, which are widely utilized for toxicity analysis and drug development. Kim produced human iPSC-derived hepatic organoids (hHOs) from human iPSCs for long-term expansion and drug testing to solve this difficulty. These organoids feature multicellular compositions, cellular polarity, hepatobiliary structures, and remarkable CYP450 activity, faithfully recapitulating metabolic clearance, CYP450-mediated drug toxicity, and metabolism [134]. The organoid was validated for hepatotoxicity caused by seven drugs and elucidated drug metabolism pathways, including those for amitriptyline, chlorpromazine, and diclofenac. Hence, human in vitro hepatic models that accurately replicate liver function are crucial for translational research and medication discovery [70,134].

### 4.3. Cardiac Organoids

Fewer new cardiovascular drugs are successful through clinical trials, mainly due to the lack of clarity about drug toxicity and potential drug effects. Thus, the demand for drug toxicity assessment and new drugs for heart disease is still steadily increasing [135,136,137].

Cardiac organoids derived from human iPSCs are now widely used for drug toxicity testing. Human iPSC-derived cardiomyocytes can elucidate the role of ceramides and mitochondrial autophagy in cardiotoxicity [138], as well as detect drug-induced cardiotoxicity for various drugs including anthracyclines [139]. However, given that 2D cell cultures cannot perfectly replicate in vivo conditions, researchers are turning to iPSCs to create 3D heart models. A simple method to generate human 3D heart microtissue from iPSCs has been proposed, which can be generated scaffold-free, cultured for extended periods, and exhibit multiple cell types found in the human heart. They preserve coordinated contractile activity for several months and respond functionally to chemotherapy drugs [140]. Additionally, iPSC-derived cardiac organoids can also be used to screen therapeutic drugs after heart transplantation, simulate the body’s rejection reaction to drugs, and explore the potential adverse reactions of drugs. For example, the potential adverse effects of immunosuppressants like calcineurin inhibitors (tacrolimus) and proliferative signaling inhibitors (sirolimus) on cardiac function post-transplantation need further investigation. Cardiac organoids from iPSC-derived cardiomyocytes, fibroblasts, and endothelial cells were created [141], and combined with single-cell RNA sequencing to examine the cardiovascular effects of these drugs, revealing favorable cardiac remodeling with proliferation signal inhibitors compared to calcineurin inhibitors [141].

However, when screening drugs for heart disease treatment, researchers often employ gene editing to create 2D in vitro models for drug screening, but few construct highly mimetic 3D organoids to replicate the internal environment for drug discovery. One of the reasons is the lack of a high-throughput screening system and homogeneous cardiac organoids, which makes drug screening based on iPSC-derived organoids a nonsense. Hence, this direction could shape the future of heart drug screening, requiring innovation and further investigation to overcome challenges in this area.

## 5. Cell Therapy Based on iPSC Technology

Although there has been a lot of excitement about using iPSCs to create tissues to treat a range of diseases, such as leukemia, PD, and liver cirrhosis, it’s still elusive how beneficial iPSC-derived cells will be in the long run for regenerative medicine. The scientific community was thrilled by the discovery of iPSC technology and the prospect of customized cell treatment. Researchers benefit from iPSCs’ ability to produce a variety of cell types on a large scale by removing barriers related to processes like apheresis, which involves removing certain blood components and returning the remaining components to the donor’s circulation in order to obtain human cells [142]. Thus, below we elaborate and summarize around this topic.

### 5.1. The Advent of iPSC-Derived DA Neurons Offers an Opportunity to Treat Neurodegenerative Disease

To date, the usage of iPSCs to model neurodegenerative diseases has become increasingly popular, and relatively few studies have been conducted using cells derived from them for treatment. Therefore, here we take PD as an example to illustrate the significance of iPSC-related cell therapy in this field.

PD, a neurodegenerative disorder affecting around 1% of the population over 60, has long relied on dopamine-replacement therapy as its primary treatment [143]. The gradual degradation of DA neurons at the significant nigra in the midbrain is the pathological hallmark of PD. Only DA neurons are harmed in PD, and the lost cells are limited by space, only in the midbrain. iPSC-derived autologous DA neurons offer an opportunity to solve the problems. Recently, most of the studies have shown that iPSC-derived DA neurons effectively restore motor function and effectively reinnervate the host brain without the potential threat of tumor formation [143,144,145,146,147,148,149,150,151,152]. For instance, researchers employed small-molecule compounds to direct human iPSCs differentiation into nigral DA neurons that closely resembled in vivo counterparts [153]. Functionally, they demonstrated autoreceptor-dependent control of dopamine release and autonomous pacemaking based on L-type voltage-dependent Ca^2+^ channels [153]. The iPSC-derived DA neurons show strong survival and axon expansion when transplanted into the striatum of 6-Hydroxydopamine hydrochloride (6-OHDA)-lesioned athymic rats. This improves motor impairments in the rat PD model [153].

Nevertheless, the application of human iPSC-based cell therapy for PD faces numerous significant obstacles. Firstly, there’s a great deal of variation in the differentiation potentials of distinct human iPSC lines, most likely due to our incomplete understanding on the reprogramming process. Secondly, there’s still much to learn about the safety of human iPSC-based cell therapy. It’s also significant to fully remove some human iPSCs from the cell mixture that are undifferentiated or have tumorigenic mutations because they have the potential to develop into cancer.

### 5.2. iPSC-Derived Hepatocytes and Immune Cells Are Effective in the Treatment of Liver Diseases

Chronic liver disease is a significant global health concern, often leading to fibrosis as a response to persistent liver injury, resulting in both healing and scar formation [154]. In certain clinical situations, hepatocyte transplantation has been indicated as a substitute for whole organ transplantation in order to stabilize and extend the lives of patients. Among them, iPSC played a crucial role as an important treatment method. iPSC-derived hepatocyte-like cells expressed hepatocyte-specific genes and proteins and displayed high levels of hepatocyte growth factor and IL-10 expression [155,156,157]. By transplanting these generated hepatocyte-like cells, abnormal liver function can be improved, and thioacetamide-induced liver fibrosis and apoptosis are dramatically alleviated [155]. Additionally, iPSCs can also differentiate into immune cells involved in the treatment of chronic liver disease. For example, macrophages derived from human iPSCs can be directed into M1 or M2 phenotypes [158]. These human iPSC-derived macrophages, in particular the M2 subtype, dramatically diminish the expression levels of fibrogenic genes and related histological markers when used to treat liver fibrosis, providing a promising cell-based method to lessen fibrosis [158]. Alternatively, iPSCs can also be directly used to treat liver disease. iPSCs naturally produce and release extracellular vesicles, which, when internalized by hepatic stellate cells, modify their profibrogenic phenotype by reducing the expression of profibrogenic markers [154]. Thus, these studies have explored the liver-regenerating potential of iPSC-derived cells, providing a solid foundation for further investigations into cell therapy as an alternative treatment for liver disorders in humans [155].

### 5.3. iPSC-Derived Cardiac Patches and Extracellular Vesicles May Be a New Approach to Treating Heart Failure

Heart failure has a significant morbidity, mortality, and cost component, making it a serious and rapidly expanding global public health concern. With approximately 5500 transplants performed annually, heart transplantation is still the major therapeutic strategy for patients with end-stage heart failure and is a challenging issue in the medical industry [141]. The challenges of heart transplantation have been the overwhelming demand and the ideal of immunosuppression-free strategies. Cardiac stem cell therapy has recently been developed and extensively investigated as an alternative therapeutic strategy for heart failure. Among them, the combination of iPSCs and drug dosage forms is a relatively novel treatment. A novel cardiac patch for cardiac function recovery was produced using 3D bioprinting technology combined with a co-culture of cardiomyocytes, fibroblasts, and endothelial cells from human iPSCs [159,160]. It was shown that rats treated with patches showed enhanced diastolic function indices determined from echocardiography, lowered left ventricular-end diastolic pressure and the time constant of left ventricular relaxation, and increased anterior wall thickness in diastole [160]. In the heart that had been infarcted, the patch enhanced the expression of genes associated with vascular endothelial growth factor, angiopoietin 1, gap junction α-1 protein, β-myosin heavy 7, and insulin growth factor-1 [160]. Alternatively, extracellular vesicles of iPSC-CMs were used for cardiac treatment [161]. To get mitochondria-rich extracellular vesicles (M-EVs), the iPSC-CMs-conditioned media was ultracentrifuged. The therapeutic benefits of M-EVs were then examined using an in vivo murine myocardial infarction model (MI) [161]. As early as 3 h following treatment, therapy with 1.0 × 10^8^/mL M-EVs significantly enhanced the contractile characteristics and intracellular adenosine triphosphate synthesis of hypoxia-injured iPSC-CMs [161]. Intramyocardial injection of M-EVs enhanced post-MI cardiac function in vivo and reduced the risk of heart failure [161,162]. Moreover, in addition to the above treatment methods, cardiac fibroblasts also have the potential to play a role in cardiac therapy. Given the possible contribution of cutaneous or human cardiac fibroblasts to cardiac repair, reprogramming these fibroblasts into iPSCs and collecting and purifying the exosomes released from these cells may lead to a therapeutic tool to support the development of cardiomyocytes [163].

### 5.4. iPSC-Derived Immune Cells Are a New Method of Tumor Immunotherapy

Adoptive cell therapy, which infuses immune cells into the patient’s body, has shown unique efficacy in the treatment of refractory malignancies. Most of the immune cells in the immune system are involved in the progression of tumors, and these cells deserve to be further studied and explored. Here, we focus on elucidating the role of some immune cells in tumor progression, including T cells, Natural killer (NK) cells, and macrophages.

#### 5.4.1. T Cells

T cells are the main force in the fight against tumors and can recognize MHC molecules on the cell surface. These immune cells can be divided into two broad classes according to the different mechanism of effect, namely, CD4^+^ T cells and CD8^+^ T cells. CD4^+^ T cells recognize MHC Class II molecules to regulate the immune system. By contrast, CD8^+^ T cells recognize MHC Class I molecules and exert cytotoxicity directly, killing alien cells. Clinically, T cell adoptive cell therapy can treat some cancers [164,165,166,167,168], however, for patients with refractory tumors, the therapeutic effect of this approach is minimal. In order to solve this problem, researchers have long found that engineered T cells are a new strategy for the treatment of recurrent and refractory tumors [169,170,171], especially chimeric antigen receptor T cells (CAR-T) [172,173,174,175,176]. Chimeric antigen receptors (CARs) are engineered proteins designed to guide T cells to target cancer cells. (The structure of the chimeric antigen receptor is shown in Figure 4 below). In preclinical models, adoptive transfer of stem-like CAR-T cells produced improved tumor control and resistance to tumor rechallenge [177]. In xenograft models, CAR-CD3^+^CD4^−^CD8^−^ T cells demonstrated efficient infiltration and tumor suppression against lung cancer genetically engineered to produce CD19, as well as antigen-specific cytotoxicity against B cell acute lymphoblastic leukemia [176]. Thus, the design of CAR-T cell therapy is feasible and effective in clinical cancer treatment.

However, T cell adoptive cell therapy and CAR T cell therapy utilizes a person’s own T lymphocytes, making it challenging for patients with progressive diseases and leukemia [178]. Additionally, the time-consuming process of generating autologous CAR T cells due to the limited availability of T lymphocytes and its inapplicability for third-party patients creates significant obstacles [178]. The fusion of iPSC and CAR technologies presents a promising avenue in oncology and significantly streamlines cell-based cancer therapy [178]. The unlimited production of CAR T cells from human iPSCs offers a compelling “off-the-shelf” CAR T cell immunotherapy approach [179]. CD8αβ^+^CAR T cells with typical characteristics can be obtained from iPSCs, and these expanded CAR-T derived from iPSCs (iPSC-CAR T cells) exhibited potent in vivo antitumor activity, extending the survival of mice with human tumor xenografts [179,180,181,182]. Furthermore, the use of gene editing technology may enable iPSC-CAR T cells to have higher anti-tumor activity. Of note, it has been demonstrated that EZH1 suppression promotes T cell maturation and differentiation from iPSCs in vitro. iPSC-T cells generated in a stroma-free, serum-free environment following silence of EZH1 exhibit strong antitumor effects when transduced with CARs both in vitro and in xenograft models [183]. It has been reported that transducing cells with genes encoding for membrane-bound intereleukin-15 (IL-15) and its receptor subunit IL-15α, as well as genetically knocking out diacylglycerol kinase, which inhibits antigen-receptor signaling, can increase the proliferation and persistency of CD8αβ^+^ cytotoxic CAR T cells in solid tumors [184]. Moreover, iPSCs produced from LMP2-specific cytotoxic T lymphocytes were treated with latent membrane protein 1 (LMP-1)-CAR to generate rejuvenated cytotoxic T lymphocytes. These engineered cells, which targeted CD19 and LMP-2 antigens, also displayed a strong tumor suppressive effect and clearly conferred a survival advantage [185]. Thus, editable and readily available therapeutically altered T cells can be produced using iPSCs, a new source with limitless potential.

#### 5.4.2. NK Cells

Compared to T cells, NK cells may be more suitable for engineering to participate in tumor therapy, because they can release perforin or granzyme directly to kill tumor cells without presenting antigens via MHC molecules [186]. Such cells play a crucial role in the immune response against viral infections and tumors [187,188,189], which have the capacity to identify and eliminate “stressed cells”, including virus-infected, allogeneic, and tumor cells. Its effector functions are governed by a balance of signals from activating and inhibitory receptors, along with cytokines such as IL-2, IL-12, IL-15, and IL-18 [190]. Activated NK cells can directly destroy tumor cells or engage in antibody-dependent cellular cytotoxicity through the CD16 membrane receptor [191].

Even though these cells are capable of identifying and eliminating tumor cells, they only make up 5% of the lymphocyte pool in peripheral blood, which implies that in order to acquire enough cells for a single therapeutic dosage, extensive donor apheresis and expensive sample processing are required [192]. iPSC, a helpful tool for creating a large number of allogeneic NK cells, effectively solves this issue [193,194,195]. It also provides a special platform for genetic modifications at the clonal level and the subsequent generation of a large number of genetically engineered NK cells derived from a standardized starting cell source. As illustrated in Figure 5. For instance, cytokine-inducible SH2-containing protein (CISH) can be easily knocked out during the differentiation phase of iPSCs, and further induction of differentiation of such iPSCs can produce standardized CISH knockout NK cells [196]. Since CARs have the potential to overcome tumor cell escape strategies and target NK cells to tumor cells carrying specific antigens, which has good implications for cancer therapy and are easily inserted into iPSC. iPSC-CAR NK cells demonstrate strong cell viability and are effective in eliminating various tumor cells [197,198,199,200]. For example, a novel CAR targeting the conserved α3 domain of MHC class I chain-related protein A (MICA) and MHC class I chain-related protein B (MICB) is incorporated into a multiplexed-engineered iPSC-derived NK cell exhibiting antigen-specific anti-tumor reactivity across an expansive library of human cancer cell lines [201]. For targeted immunotherapy, the improved human iPSC-based approach may provide a workable way to produce CAR NK cells with an immunological memory-like nature.

However, CAR also has deficiencies in cancer treatment. The capacity of CAR-NK cells to interact with their target is hampered by the decreased tumor antigen density caused by the transfer of the CAR cognate antigen from the tumor to NK cells, as demonstrated by a recent study [202]. A dual-CAR system might counteract the phenomenon. The researchers use NK self-recognizing inhibitory CAR, which controls NK cells by sending them a “don’t kill me” signal, in addition to activating CAR against the cognate tumor antigen [203].

#### 5.4.3. Macrophages

Macrophages play also an important role in cancer development and metastasis [204,205,206,207,208,209,210]. There are significant challenges in expanding the use of CAR T therapy across various cancer types, primarily due to its limited effectiveness caused by the intricate tumor microenvironment. It’s possible that manipulating macrophages to modify the tumor immune microenvironment is a more effective tumor therapeutic strategy. iPSC-derived macrophage cells are a fantastic source of macrophage cells because immortalized macrophage cell lines are not suitable for use in clinical settings and primary macrophages produced from bone marrow or PBMCs are not effectively engineered [211]. By producing CAR-expressing macrophage cells from iPSCs, the procedure has advanced significantly [212]. These modified macrophages have been illustrated to have better functions, including higher cytokine production and expression, polarization toward a pro-inflammatory/anti-tumor state, greater tumor cell phagocytosis, and in vivo anticancer effectiveness (Figure 6). Hence, macrophages derived from iPSCs have potential applications in cancer treatment.

## 6. Conclusions and Perspectives

The choice of reprogramming method depends on factors such as efficiency and safety, which are critical considerations depending on the purpose of generating iPSCs. For basic research studies where safety is less of a concern, the choice of method is more flexible. However, for clinical applications, safety is paramount, regardless of efficiency. So far, iPSC research leans more towards applications rather than reprogramming methods. This suggests that reprogramming methods are still in the early stages of iPSC development. While modern advanced technology may lead to breakthroughs in reprogramming methods, it’s essential to consider the balance between research costs and benefits. Since the initial description of diseased iPSCs, models have evolved from simple differentiated 2D cultures of single lineage cells with cellular-level outputs to more complex 3D organoids and in vivo chimeras. These advancements now incorporate interactions at the cell-tissue and tissue-organ levels into disease models. The emergence of human-specific modalities that require in vitro models of human cells and molecular biology has accelerated the humanization of drug discovery [213]. Perhaps, continuous improvements in iPSC differentiation protocols and chimeric models have the potential to revolutionize various fields, including infectious diseases, oncology, and tissue transplantation.

There’s a wide variety of cells that can be derived from iPSCs, and the advanced methods used in this process hold great promise. Autologous iPSC-derived cells are considered the safest in terms of immune rejection. While many iPSC-derived cells have shown therapeutic effects on diseases, their application remains mostly in the laboratory. Clinical transformation takes time and incurs relatively high costs. During the differentiation of iPSCs into specific cells, mutations can occur that interfere with treatment, leading to cases where patients have had to discontinue surgeries. Additionally, there’s a shortage of allogeneic iPSCs available for use. A significant challenge is ensuring that stored cells can serve a wide range of individuals. Many institutions are building inventories of iPSCs from healthy donors, specifically generated from the blood of human leukocyte antigen homozygous donors to minimize the risk of tissue rejection following transplantation. Once the inventory is built, it will greatly advance human medicine.

To date, site-specific genome editing tools like CRISPR/Cas9 have enabled the correction of target gene mutations. The combination of iPSC technology and CRISPR/Cas9 gene editing may be able to better improve the treatment of major diseases such as cancer, thereby ushering in a new era of treatment. Furthermore, chip technology has gained popularity, allowing for controllable culture conditions, diverse stimulations, and continuous parameter read-outs. If a new multi-organ chip based on human iPSCs is developed, it may effectively address the major problems such as organoid heterogeneity. Therefore, generating iPSC-derived organoids on chips or exploring interactions between iPSC-derived organoids and other organoids via chips may represent a promising avenue for future research.

## Figures and Tables

**Figure 1 ijms-25-02680-f001:**
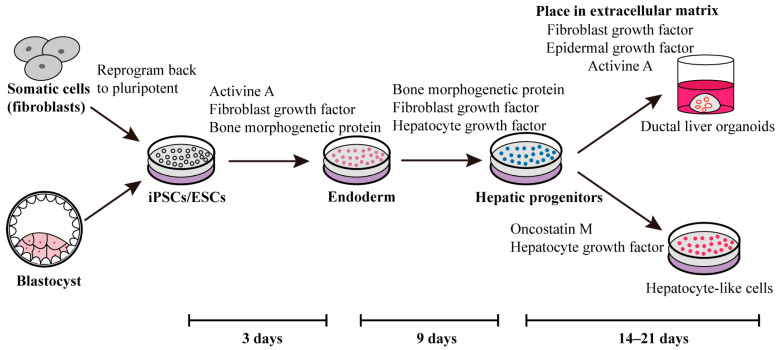
One approach involves employing self-renewal factors to reprogram fibroblasts or blastocysts into iPSCs or ESCs. During the initial three days, activin A, fibroblast growth factor, and bone morphogenetic protein are introduced to prompt stem cell differentiation towards the endoderm lineage. Subsequently, over the following six days, fibroblast growth factor, bone morphogenetic protein, and hepatocyte growth factor are administered to facilitate the differentiation of hepatic progenitors. Ultimately, this process yields ducted liver organoids or hepatocyte-like cells, achieved through the influence of various stimulating factors [66,67,68,69,70].

**Figure 2 ijms-25-02680-f002:**
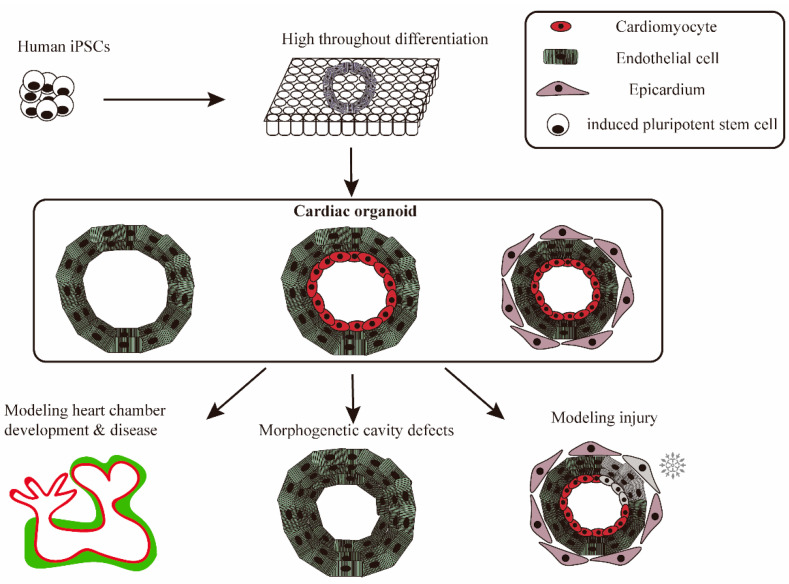
iPSC-derived cardiac organoids mimics heart disease and physiological and pathological processes. iPSCs are widely differentiated into cardiac organoids in pore plates, which are used to simulate heart disease. Cardiac organoids with different cellular compositions are used for the study and exploration of different disease mechanisms or normal physiological processes, including injury, heart chamber development, and morphogenetic cavity defects.

**Figure 3 ijms-25-02680-f003:**
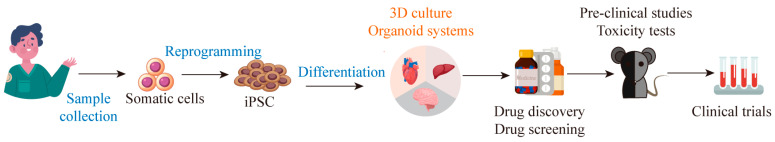
iPSC-derived organoids can be used for drug screening. iPSCs derived from somatic cells can differentiate into 3D organoids, which can provide a disease model that replicates disease pathophysiology at the tissue and organ level for drug screening.

**Figure 4 ijms-25-02680-f004:**
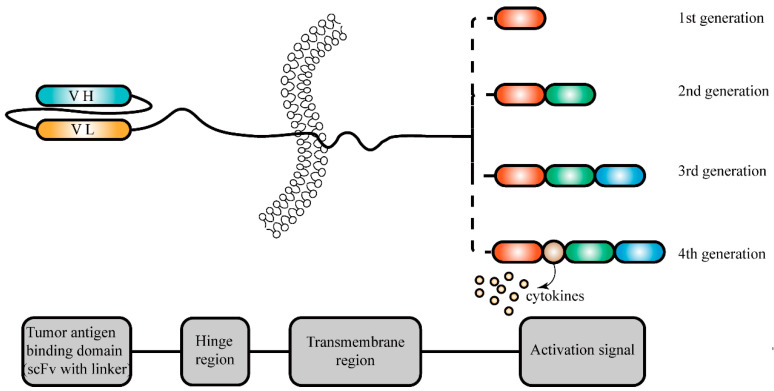
Structural composition of chimeric antigen receptors for tumor treatment. Chimeric antigen receptors (CARs) are engineered proteins designed to guide T cells to target cancer cells. Typically, a CAR comprises an antigen recognition domain from a monoclonal antibody (mAb), a hinge region, a transmembrane domain, a costimulatory domain, and a T-cell receptor activation domain. The antigen recognition domain usually contains a single-chain variable fragment (scFv) derived from antibodies, enabling specific recognition of tumor antigens on cancer cells. Different generations of chimeric antigen receptors result in different activation signal regions.

**Figure 5 ijms-25-02680-f005:**
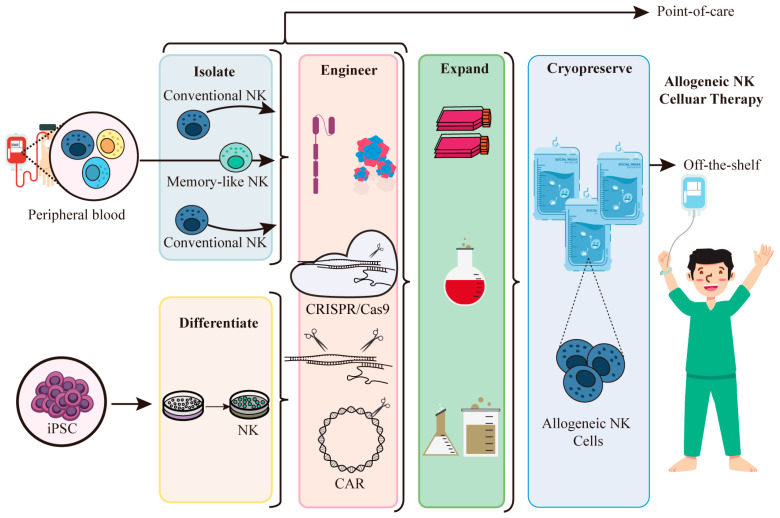
iPSC-derived NK cells are used for allogeneic cell therapy. NK cells can be isolated from the peripheral blood, but they represent only a small fraction of the peripheral blood lymphocyte pool. This requires extensive donor beading and extensive sample handling. iPSC is a valuable resource for producing NK cells for allogeneic therapy. Acquired NK cells can be genetically modified at the clonal level, as is done with NK cells isolated from peripheral blood, and then a large number of genetically engineered NK cells can be obtained from a standardized source of initial cells for allogeneic cellular therapy.

**Figure 6 ijms-25-02680-f006:**
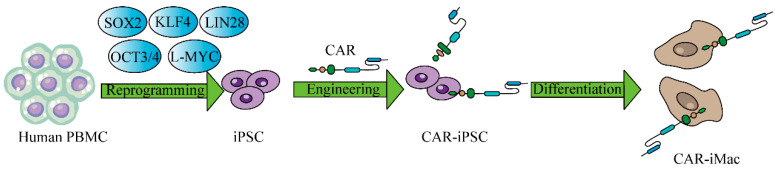
CAR-expressing iPSCs can differentiate into CAR-macrophage cells. The overview of deriving CAR-iMacs from CAR-iPSCs.

**Table 1 ijms-25-02680-t001:** Different reprogramming methods.

Method	Type	Efficiency	Preparation of Materials	Delivery Procedure	Remove of Exogenous Factors
Sendai virus [12,13,27]	Virus	+++	Difficult	Easy	Easy
Adenovirus [11,28]	Virus	+	Difficult	Moderate	Moderate
Plasmid [15]	DNA	+	Easy	Difficult	Moderate
Episomal plasmid [13,29]	DNA	++	Easy	Easy	Moderate
PiggyBac transposon [30,31]	DNA	++	Difficult	Easy	Moderate
mRNA [17,19]	RNA	+++	Difficult	Difficult	Easy

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
