# Peer review of "Application Prospect of Induced Pluripotent Stem Cells in Organoids and Cell Therapy"

_ijms, 2024, doi:10.3390/ijms25052680_

Round 1
Reviewer 1 Report
Comments and Suggestions for Authors
The article is devoted to induced pluripotent stem cells. The authors have tried to describe all aspects of this technology. Unfortunately, this is where the imperfection of the review lies. Each chapter could become a separate large article. Currently the review resembles a patchwork quilt - there are many parts in the text (establishing transgene-free iPSCs, various types of organoids, cancer organoids, iPSC-derived cells for therapeutic purposes, including CAR-T cells. Unfortunately, the desire to talk about everything has led to the fact that the review article looks a little superficial, it does not feel the opinion of the authors. I would like to advise the authors to reduce the number of topics discussed in the review, but to make the text of the article more analytical and in-depth.
Author Response
Response: We highly appreciate your thoughtful comments. Our aim was to provide a comprehensive introduction and summary of non-integrative reprogramming methods of iPSC technology, and deliver a comprehensive description of iPSC-derived organoids, spanning the realms of the nervous system, cardiovascular system, and oncology, as well as systematically elucidate recent advancements in iPSC-related cell therapies. According to your suggestion, we have now made striking effort to expand the analysis and describe in-depth in the conclusions and perspectives section, as shown in the revised manuscript.
Reviewer 2 Report
Comments and Suggestions for Authors
First of all, I would like to thank the authors for this review, which is quite enjoyable to read. The topic compiled by the authors is undoubtedly valuable for its focus on applications in tissue engineering that go beyond classical cell culture practices, a subject we increasingly encounter in today's world. The compilation of applications in this field is unquestionably beneficial due to the aim of moving beyond the current era in tissue engineering applications.
The study has been comprehensively addressed in general terms. While table and figure descriptions seem adequate, the lack of references is particularly noticeable within the tables. For example, Table 1, which provides a summary of different reprogramming methods, does not contain any references. Although some explanations are provided in the text, it is essential for the reader to have access to these sources. Therefore, the information in the tables needs to be reinforced with references. Similarly, especially in figures like Fig. 1, which provides an overview of the topic and utilizes information sources in the captions of detailed figures, the sources of information used - if taken from an article (e.g., 3 days - 9 days, 14-21 days info) - should be included.
Fig. 3 is presented as a summary figure. The caption mentions that "iPSCs derived from somatic cells can differentiate into 2D cells and 3D organoids, both of which can provide a disease model that faithfully replicates disease pathophysiology at the tissue and organ level for drug screening." However, under the subsequent heading, there is the expression "Organoids present some advantages over traditional 2D cultures," which is correct. It would be beneficial to clarify the statement in the Fig caption regarding 2D cells showing significant pathology at the tissue and organ levels.
On page 18, line 766: Could you please explain how multi-organ chip technology contributes to organoid heterogeneity? While its microfluidic structure allows control over differentiation parameters through both laminar flow and the effect of controllable shear stress, which is certainly accurate, what is the reason for expecting it to reduce heterogeneity, especially in systems derived from multi-organ chips? This involves multiple differentiation and control parameters. While acknowledging the benefits of this technology for single-organ differentiation, how do you envision these systems, which provide controllable parameters for differentiation in a single organ, reducing heterogeneity when applied to multiple organs?
Author Response
Reviewer#2: First of all, I would like to thank the authors for this review, which is quite enjoyable to read. The topic compiled by the authors is undoubtedly valuable for its focus on applications in tissue engineering that go beyond classical cell culture practices, a subject we increasingly encounter in today's world. The compilation of applications in this field is unquestionably beneficial due to the aim of moving beyond the current era in tissue engineering applications.
Response: We highly appreciate your thoughtful comments, which have provided great opportunity to improve our manuscript.
The study has been comprehensively addressed in general terms. While table and figure descriptions seem adequate, the lack of references is particularly noticeable within the tables. For example, Table 1, which provides a summary of different reprogramming methods, does not contain any references. Although some explanations are provided in the text, it is essential for the reader to have access to these sources. Therefore, the information in the tables needs to be reinforced with references. Similarly, especially in figures like Fig. 1, which provides an overview of the topic and utilizes information sources in the captions of detailed figures, the sources of information used - if taken from an article (e.g., 3 days - 9 days, 14-21 days info) - should be included.
Response: Thank you very much for your helpful comment. According to your suggestion, the required references have now been added on Page 3-4 in Table 1. In addition, the corresponding information sources and references for Figure 1 have now been added in Line 215 on Page 5.
Fig.3 is presented as a summary figure. The caption mentions that "iPSCs derived from somatic cells can differentiate into 2D cells and 3D organoids, both of which can provide a disease model that faithfully replicates disease pathophysiology at the tissue and organ level for drug screening." However, under the subsequent heading, there is the expression "Organoids present some advantages over traditional 2D cultures," which is correct. It would be beneficial to clarify the statement in the Fig caption regarding 2D cells showing significant pathology at the tissue and organ levels.
Response: Thank you very much for your comment. Both 2D cells and 3D organoids generated from iPSCs have certain capacity to mimic physiological and pathological conditions. We have illustrated the differences between iPSC-derived 3D organoids and 2D cultures in Line 403-409 on Page 9, as well as in Line 419-427 on Page 10. Additionally, we have modified Figure 3 and edited the correspond figure legend in Line 415-417 on Page 10.
On page 18, line 766: Could you please explain how multi-organ chip technology contributes to organoid heterogeneity? While its microfluidic structure allows control over differentiation parameters through both laminar flow and the effect of controllable shear stress, which is certainly accurate, what is the reason for expecting it to reduce heterogeneity, especially in systems derived from multi-organ chips? This involves multiple differentiation and control parameters. While acknowledging the benefits of this technology for single-organ differentiation, how do you envision these systems. which provide controllable parameters for differentiation in a single organ, reducing heterogeneity when applied to multiple organs?
Response: Your valuable comment is greatly appreciated. Indeed, organogenesis is deemed to be a complicated process. Organ maturation is facilitated by intertissue communication in pluripotent stem cell-produced multi-organ systems. Multi-organ chip technology leading to organoid heterogeneity was mentioned in Line 243-257 on Page 6. Unfortunately, the differentiation parameters for providing individual organoids in multi-organ chips have not been reported. Herein, we proposed a possible way for the co-formation of two kinds of organoid to solve the heterogeneity in multi-organ chip. We have modified and polished the content in Line 743-748 on Page 16-17.
Reviewer 3 Report
Comments and Suggestions for Authors
Zhang and coworkers describe application prospect of iPSCs in organoids and cell therapy such as brain, liver, heart, and cancer tissues. The review manuscript is well written and covered with recent researches using a variety of iPSC-derived cells. For publication, there are some minor concerns as follows.
3.4. Derivation of cancer organoids
1) Matano and coworkers demonstrated modeling colorectal cancer using CRISR-Cas9- mediated engineering of human intestinal organoids (Nat. Med. 2015, 21, 256-262). The content of the research is focused on tissue stem cells, not on iPSCs. Please cite the paper and discuss comparatively.
2) The authors describe retinal organoids in this section. Concerning eye organoids derived from human iPSCs, the authors should cite the following paper (Hayashi et al. Nature, 2016, 531, 376-380) and describe transplantation of retina derived from human iPSCs.
4.1. Neural organoids
3) Recently, tissue-derived fetal brain organoids (FeBOs) were established (Hendriks et al. Cell 2024, 187, 1-21). Though the organoids are derived from fetal tissue stem cells, please describe as the recent research with citation.
4) Recently, Okano group published the paper concerning clinical trial in ALS with ropinirole as a drug candidate indentified by iPSC drug discovery (Morimoto et al. Cell Stem Cell 2023, 30, 766-780). Please add the paper for discussion in your manuscript.
5) Page11, line475: acetaminophen -> APAP.
6) Page16, line673: [167], [168], [169] -> [167-169].
7) Page18, line764: ... a new era of treatment -> please add comma.
References
77: There are no author's names.
Author Response
Reviewer#3: Zhang and coworkers describe application prospect of iPSCs in organoids and cell therapy such as brain, liver, heart, and cancer tissues. The review manuscript is well written and covered with recent researches using a variety of iPSC-derived cells. For publication, there are some minor concerns as follows.
Response: We highly appreciate your encouraging comments.
3.4. Derivation of cancer organoids
1) Matano and coworkers demonstrated modeling colorectal cancer using CRISR-Cas9- mediated engineering of human intestinal organoids (Nat. Med. 2015, 21, 256-262). The content of the research is focused on tissue stem cells, not on iPSCs. Please cite the paper and discuss comparatively.
Response: Thank you very much for your helpful comment. According to your advice, we have now cited this paper and discussed comparatively in Line 361-368 on Page 8-9.
2)The authors describe retinal organoids in this section. Concerning eye organoids derived from human iPSCs, the authors should cite the following paper (Hayashi et al. Nature, 2016, 531, 376-380) and describe transplantation of retina derived from human iPSCs.
Response: Thank you very much for your valuable comment. According to your suggestion, we have now cited the paper and provided a through description in Line 381-385 on Page 9.
4.1. Neural organoids
3)Recently, tissue-derived fetal brain organoids (FeBOs) were established (Hendriks et al. Cell 2024, 187, 1-21). Though the organoids are derived from fetal tissue stem cells, please describe as the recent research with citation.
Response: Your helpful comment is highly appreciated. According to your suggestion, we have now cited the paper and provided a through description in Line 451-460 on Page 10.
4) Recently, Okano group published the paper concerning clinical trial in ALS with ropinirole as a drug candidate indentified by iPSC drug discovery (Morimoto et al. Cell Stem Cell 2023, 30, 766-780). Please add the paper for discussion in your manuscript.
Action: We highly appreciate your thoughtful comment. According to your suggestion, we have now cited this paper and discussed it in Line 436-441 on Page 10.
5)Page11, line475: acetaminophen -> APAP.
Action: Thanks for your suggestion. We have corrected it in Line 479 on Page11.
6)Page16, line673: [167], [168], [169] -> [167-169].
Action: Thank you so much for your helpful suggestion. We have now corrected it in Line 662 on Page 14.
7)Page18, line764: ... a new era of treatment -> please add comma. References 77. There are no author's names.
Action: Your helpful comment is highly appreciated. According to your suggestion, we have now corrected it in Line 745 on Page 16. In addition, References 77 has now been corrected on in Line 993-996 on Page 22.
Round 2
Reviewer 1 Report
Comments and Suggestions for Authors
no further concern